# Modeling the Effect of Cannabinoid Exposure During Human Neurodevelopment Using Bidimensional and Tridimensional Cultures

**DOI:** 10.3390/cells14020070

**Published:** 2025-01-07

**Authors:** Enrique Estudillo, Jorge Iván Castillo-Arellano, Emilio Martínez, Edgar Rangel-López, Adolfo López-Ornelas, Roxana Magaña-Maldonado, Laura Adalid-Peralta, Iván Velasco, Itzel Escobedo-Ávila

**Affiliations:** 1Laboratorio de Reprogramación Celular, Instituto Nacional de Neurología y Neurocirugía Manuel Velasco Suárez, Mexico City 14269, Mexico; jorge.ivan@ciencias.unam.mx (J.I.C.-A.); emimartinez2424@ciencias.unam.mx (E.M.); raledg@hotmail.com (E.R.-L.); rmagana@innn.edu.mx (R.M.-M.); laura.adalid@innn.edu.mx (L.A.-P.); ivelasco@ifc.unam.mx (I.V.); 2Facultad de Ciencias, Universidad Nacional Autónoma de México, Mexico City 04510, Mexico; 3División de Investigación, Hospital Juárez de México, Mexico City 07760, Mexico; adolfolopezmd@gmail.com; 4Hospital Nacional Homeopático, Hospitales Federales de Referencia, Mexico City 06800, Mexico; 5Instituto de Fisiología Celular-Neurociencias, Universidad Nacional Autónoma de México, Mexico City 04510, Mexico

**Keywords:** drugs, pregnancy, stem cells, cell reprogramming, neurogenesis, embryonic development

## Abstract

Our knowledge about the consumption of cannabinoids during pregnancy lacks consistent evidence to determine whether it compromises neurodevelopment. Addressing this task is challenging and complex since pregnant women display multiple confounding factors that make it difficult to identify the real effect of cannabinoids’ consumption. Recent studies shed light on this issue by using pluripotent stem cells of human origin, which can recapitulate human neurodevelopment. These revolutionary platforms allow studying how exogenous cannabinoids could alter human neurodevelopment without ethical concerns and confounding factors. Here, we review the information to date on the clinical studies about the impact of exogenous cannabinoid consumption on human brain development and how exogenous cannabinoids alter nervous system development in humans using cultured pluripotent stem cells as 2D and 3D platforms to recapitulate brain development.

## 1. Introduction

The use of cannabinoids during pregnancy has seen a noticeable increase in recent years, influenced by factors that extend beyond managing morning sickness. Many pregnant women turn to cannabinoids to help alleviate stress, insomnia, or pain. This growing trend may also stem from the widespread perception of cannabis as a natural or relatively safe remedy. Moreover, its greater availability due to legalization in various regions has contributed to its accessibility [1]. This social phenomenon raises concerns regarding to the impact of marijuana consumption on fetal development. The main component of marijuana, tetrahydrocannabinol (THC), is a cannabinoid that can cross the placental barrier and reach the developing fetus. Interestingly, during human development, the developing nervous system expresses the endocannabinoid receptors CB1 and CB2 [2,3], which enable responses to circulating endocannabinoids. These receptors play critical roles in synaptic regulation, axonal guidance, and neuronal differentiation [4]. Their presence highlights the system’s susceptibility to external cannabinoids while emphasizing their essential function in supporting normal neurodevelopment. This evidence strongly suggests that exposure to cannabinoids such as THC during pregnancy could impair the development of the human brain.

Unfortunately, retrospective analyses of fetuses exposed to exogenous cannabinoids usually present confounding factors such as exposure to alcohol or cigarettes that make hard to elucidate whether cannabinoid consumption is the main cause of neurological impairments derived from alterations in fetal development [5,6,7,8]. Embryonic stem cells (ESCs) and induced pluripotent stem cells (iPSCs) are promising tools to overcome this drawback since they derive from the inner mass of blastocysts and the cell reprogramming of somatic cells by inducing in them the ectopic expression of pluripotency markers, respectively. Neural differentiation of ESCs and iPSCs model the specific cell types and cell interactions that occur within some brain structures, offering valuable insights into their formation and functionality. This property can provide novel insights regarding the impact of cannabinoid consumption during the development of multiple structures of the human brain.

In this review, we analyze the most recent evidence about the roles of cannabinoids during brain development, the impact of cannabinoid consumption during human neurodevelopment from clinical studies, and the effect of cannabinoids on bidimensional and tridimensional neuronal cultures derived from either ESCs or iPSCs to provide an updated and more complete interpretation about the effect of exogenous cannabinoids on human neurodevelopment. As a starting point, it is necessary to explain briefly the endocannabinoid system in the adult stages.

## 2. General Properties of Cannabinoids in the Mature Nervous System

The term “cannabinoid” is commonly used to define any molecule that is capable of binding to the cannabinoid receptors present in brain and body cells, regardless of its structure or its origin, producing effects like those produced by the phytocannabinoids present in the cannabis plant. Cannabinoids are classified based on their source as endocannabinoids (eCBs), phytocannabinoids, and synthetic cannabinoids (SCs) (Figure 1) [9,10]. eCBs are the natural agonists of the CB1 and CB2 cannabinoid receptors that, together with the enzymes required for the synthesis and degradation of eCBs, form the endocannabinoid system. These receptors are G protein-coupled receptors and regulate cognitive functions, neuronal development, pain, food intake, and the reward and addiction system [11,12].

### 2.1. Cannabinoid Receptors

CB1 is generally expressed in all neurons of the nervous system. However, it is most abundant in areas of the brain such as the basal ganglia, hypothalamus, putamen, caudate, amygdala, hippocampus, cerebellum, cortex, midbrain, spinal cord, substantia nigra, and thalamus [13,14,15]. CB1 regulates several physiological functions such as cognition, emotion, pain, energy balance, feeding, and neuroinflammation [13,16,17,18].

CB1 is preferentially found in presynaptic axon terminals of inhibitory GABAergic and glutamatergic interneurons, while eCBs are released from postsynaptic neurons and act on presynaptic CB1 to produce a short- and long-term suppression of neurotransmitter release and the modulation of neuronal activity and network function [19,20]. Moreover, eCBs can enter the extracellular space, activating CB1 receptors in nearby astrocytes. Finally, eCBs are also capable of reaching CB1 receptors in mitochondria, where their activation has a role in regulating ATP production and respiration [19,20]. Recent evidence describes that CB1 mRNA is deregulated in diseases such as cancer, schizophrenia, obesity, diabetes, and Parkinson’s and Huntington’s diseases as a result of exposure to phytocannabinoids and SCs [16,21]. Therefore, CB1 antagonists have been proposed for reducing obesity and treating Parkinson’s disease [22].

CB2 is also located in the brain, in neurons, glia, and endothelial cells, although at a lower density than CB1 [16]. There are two isoforms of CB2: CB2A, which has been found in testicles and at lower expression in some regions of the brain such as the cortex, amygdala, hippocampus, caudate, putamen, cerebellum, and nucleus accumbens; and CB2B, which is located mainly in leukocytes and the spleen [23,24]. Recent evidence suggests that CB2 may have an important modulatory role in brain regions related to drug addiction [25]. CB1 and CB2 have been identified in other peripheral tissues such as the liver, heart, adipose tissue, lungs, skeletal muscle, stomach, and reproductive organs [16]. CB2 regulates protein kinase signaling pathways involved in the survival and proliferation of progenitor cells, although the expression of this receptor decreases during neuronal differentiation. An essential role of this receptor has been its expression and regulation in cells of the immune system, as well as in bone remodeling [16,26].

Compared to CB1, CB2 shows lower expression levels than CB1, which suggests that CB2 does not have an essential psychoactive role under normal conditions. However, CB2 is inducible since its brain expression increases in anxiety, addiction, or inflammation, suggesting a relationship between CB2 expression and some neurological and psychiatric conditions [25]. Recent studies in murine models lacking CB1 in a kainate model of temporal lobe epilepsy demonstrated that the absence of CB1 produced an increase in seizures. On the other hand, the absence of CB2 showed memory impairment and schizophrenia-like behaviors [23,27,28]. The double-knockout mice for CB1 and CB2 showed that 38% of the animals died prematurely and 80% were epileptic [27]. Additionally, Δ9-tetrahydrocannabinol (THC) produced a decrease in extracellular dopamine in the nucleus accumbens (NAc) of WT and CB1 knockout (KO) mice, while in CB2 KO mice, it had the opposite effect by increasing extracellular dopamine [22].

### 2.2. Types of Cannabinoids

The endocannabinoids N-arachidonoylethanolamine, also known as Anandamide (AEA), and 2-arachidonoylglycerol (2-AG) were the first described endogenous ligands for cannabinoid receptors. AEA and 2-AG showed physiological properties like phytocannabinoids and SCs [9]. Unlike neurotransmitters, eCBs have a short half-life and are synthesized on demand, since they cannot be stored in synaptic vesicles. eCBs are metabolized by two enzymes; mainly, fatty acid amide hydrolase (FAAH), which degrades the AEA, and monoacylglycerol lipase (MAGL), which degrades 2-AG [13,19,29].

In the nervous system, eCBs act as retrograde neurotransmitters, since a postsynaptic neuron synthetizes and releases them on the presynaptic terminal, where they bind to the cannabinoid receptors present in the presynaptic neuron. The activation of the cannabinoid receptor leads to a hyperpolarization of the neuron, consequently inhibiting the release of neurotransmitters such as γ-aminobutyric acid (GABA) and glutamate [16,29,30]. eCB retrograde signaling is the regulatory mechanism mediating short- and long-term plasticity in both excitatory and inhibitory synapses. This regulation takes place in the cerebellum, dentate gyrus, hippocampus, neocortex, and substantia nigra [16,31].

Once the eCB binds its receptor, there is a presynaptic inhibition of the Ca^2+^ flow, promoted by the inhibition of the CaV-type calcium channels and the activation of the G Protein-Coupled, Inward-Rectifying K^+^ (GIRK) channels [16,31]. Additionally, the activation of the GIRK channels reduces calcium flow due to an increase in K+ conductance, producing hyperpolarization (Figure 2) [16,31]. This Ca^2+^ flow is responsible for a balance between the processes of Ca^2+^-dependent activation, such as muscle contraction and the release of neurotransmitters [16,31]. eCBs act as a brake mechanism to regulate the transient activation of synapses when neurons are stimulated beyond a given threshold [19].

In addition to AEA and 2-AG, there are other eCBs with different affinities to cannabinoid receptors, such as docosahexaenoylethanolamine, linoleoylethanolamide, oleoylethanolamide, or palmitaylethanolamide. These are synthesized through alternative biochemical routes [13,17]. They can also activate other non-cannabinoid receptors, such as GPR55, TRPV1, PPARα, PPARγ, and GPR119 [19,34].

Phytocannabinoids come from the plant Cannabis sativa, and the phytocannabinoid tetrahydrocannabinol (Δ9-THC) is the most famous and well-known cannabinoid, described in 1964, and its study led to the discovery of the endocannabinoid system [11,35]. This molecule has a partial agonism on cannabinoid receptors and consequently displays a dose-dependent biphasic effect similar to other phytocannabinoids such as cannabidiol (CBD), ECBs, and SCs [29]. Currently, the Food and Drug Administration (FDA) has authorized two synthetic Δ9-THC formulations, dronabinol (Marinol^®^, Syndros^®^) and nabilone (Cesamet™), for the treatment of chronic pain, chemotherapy-induced vomiting, and nausea-like symptoms and one CBD extract for childhood epilepsy (Epidiolex^®^) [11].

Despite the high worldwide consumption of Δ9-THC, especially in highly developed countries, no lethal doses or deaths have been associated with overdose cases. However, persistent paranoia or panic attacks are the main adverse effects of Δ9-THC consumption [29]. Nonetheless, chronic cannabis use has been associated to problems during pregnancy and the prenatal stage, such as difficulties with embryo implantation, smaller size of fetuses, preterm birth, low birth weight, admission to neonatal intensive care, and alterations of brain development with long-lasting effects on cognitive functions [29,36].

Synthetic cannabinoids are part of the largest class of drugs detected in the USA, at 190, and around the world at 280, between 2008–2018. Many SCs are currently included in the FDA’s Schedule I of the Controlled Substances Act signed by President Obama in 2012. However, SCs are constantly being created and several of these new compounds cannot be classified on Schedule I, and SCs are not part of the standard drug tests [37]. Among the effects reported with SC poisoning are some that can be lethal related to the cardiovascular system, such as ventricular fibrillation, tachycardia, myocardial infarction, coronary arterial thrombosis, and sudden cardiac death; hematological, such as intracranial hemorrhage, immune thrombocytopenia, and coagulopathy; neurological, such as tremor, dizziness, altered mental status, acute ischemic infarction, and seizure; psychiatric, such as anxiety, suicidal ideation, depersonalization, paranoia, psychosis, delirium, and hallucinations; and renal, such as acute tubular necrosis and acute kidney injury, among others [37] (United Nations Office on Drugs and Crime 2019).

## 3. Cannabinoid Signaling During Neurodevelopment

During brain development, multiple signaling pathways must be present to connect the different cell groups that will form the mature nervous system. Cell-to-cell contacts modulate this complex intercommunication through a gradient of different molecules called morphogens. Morphogens are secreted during the proliferation of neural progenitor cells and determine the programmed movement and distribution of neural and glial progenies resulting from the proliferation, distribution, and migration of these cell lineages during the formation of the cerebral cortex, cerebellum, and spinal cord [38,39]. Several lipid-related molecules, such as sphingolipids, glycolipids, phospholipids, thromboxanes, and prostaglandins, constitute these bioactive signals and are crucial for synaptic differentiation and plasticity [40].

### 3.1. Endocannabinoid Signaling in Neurodevelopment

Endocannabinoids exhibit this type of lipid-based structure and are currently recognized as neurotransmitters involved in signal transduction, modulating various cellular processes such as neurodevelopment and cell fate, synaptic transmission, cognitive processes, immune response, and energy production. It has been described that not only do endocannabinoids have a continuous effect on cell–cell communication and various intracellular signaling pathways but also are actively involved in the early development of the prenatal and perinatal brain by modulating the network of neuronal connections and the proper function of synapses even in the postnatal and mature stages of the nervous system [40,41].

In the first stages of neuronal differentiation, endocannabinoids act in an autocrine manner, a mechanism that contrasts with their paracrine function of retrograde signaling that they exert in the synaptic process in the adult organism. At these developmental stages, the location of cannabinoid receptors in the membrane remains constant, but there are differences in the intracellular location and distribution of enzymes and metabolic precursors that determine the production, transport, and degradation of endocannabinoids; this has been reported for diacylglycerol lipase-α (DAGLα), which switches from an axonal location to the postsynaptic region during differentiation and the initiation of synaptic neurotransmission [42]. DAGLα is located in specialized membrane structures near areas where the CB1 receptor is concentrated, transforming these regions into active centers of signal transduction. Following neuronal polarization and along the directional growth of the axon, 2-AG is not degraded, but there is an increase in the enzymatic turnover of MAGL at the motile sites of neurites, particularly in the growth cones. Thus, increased MAGL levels at axonal terminals prevent premature activation of cannabinoid receptors during axonal growth and guidance [43,44,45].

Similarly, axons in the thalamocortical region exhibit increased MAGL levels, which create CB1-enriched axonal pathways during corticothalamic-thalamocortical contact, defining pathways for the migration of cortical interneurons expressing high levels of these receptors [44,46]. Neural progenitor cells are precursors of cortical neurons and actively act as scaffolds for radial migration of these cells. High levels of FAAH and MAGL enzymes have been detected in cultures of neural progenitor cells, suggesting that inhibition of the activity of enzymes that degrade AEA and 2-AG is a critical step that enables endocannabinoid-induced signaling during nervous system development [45,47,48]. The simultaneous expression of DAGL and CB2 has been detected in neural progenitor cells, suggesting autocrine stimulation of these cells induced by endocannabinoids, allowing the asymmetric proliferation, modulation of the cell cycle, and prolonged migration of cell lineages resulting from this cell division [49,50].

CB1 has also been reported to be involved in the activation of some intracellular signaling pathways, such as the PI3K-AKT-mTOR pathway, in association with several mitogen-activated protein kinases (MAPK) such as ERK, p38, and JNK. These interactions are involved in the cytoskeletal instability that defines the morphological phenotypes of neurons during their development into adult organisms [51]. On the other hand, activation of CB1 by 2-AG may act as a key effector of neurotrophin signaling that promotes neurite outgrowth in a CB1-dependent manner [52]. Galve-Roperh et al. (2013) [53] demonstrated that endocannabinoids control the size of the population of neural progenitor cells in the developing forebrain and determine the cell fate of differentiation to a neuronal or astroglial phenotype in the cerebral cortex and hippocampus. These neurogenic fate decisions coincide with a CB2-to-CB1 switch in postmitotic neurons and in cortical neuronal projections, leading to the cessation of proliferation and simultaneously triggers autocrine 2-AG signaling [50,54].

The activities of CB1, CB2, TRPV, and GPR55 are also involved in neurite elongation in the developing cerebrum as they are localized at the end of axons and trigger both autocrine and paracrine stimulations by modulating signals evoked by Ca^2+^ influx into axonal growth cones and inducing remarkable reorganizations of cytoskeletal conformation involving multiple signaling molecules such as RAS, GTPases, β-catenin, and PI3K-Akt pathways [55,56,57,58].

### 3.2. Role of Cannabinoids in Axon Elongation and Synaptogenesis

After newly differentiated neurons have migrated and become established, they begin to elongate their axons towards their target sites, allowing the formation of functional neuronal circuits in the adult organism [59], through a structure at the tip of the axon called the growth cone. This sensorimotor and transient structure determines the direction of growth depending on the guidance molecules located in the microdomains of the brain [60].

The role of endocannabinoids as regulators of axonal guidance has been suggested by the presence of CB1 in the growth cone and in the developing axonal tracts [61,62]. CB1 is not only involved in axonal guidance, as its expression increases throughout the process of neuronal differentiation, but it is also detectable immediately after neuronal progenitors exit the mitotic phase of their cell cycle [63,64]. On the other hand, CB2 activation is involved in growth cone architecture [43]. Moreover, not only has the presence of CB1/CB2 in the growth cone been demonstrated, but also an increased expression of enzymes involved in the synthesis and degradation of AEA and 2-AG [43,44,61]. Increasing research on eCBs has recognized that activation of their receptors during axonal guidance elicits different responses depending on the neuron type, embryonic stage, experimental conditions, and agonists used [65].

When using models with immortalized neuronal lineages, some authors report contradictory results, which could be due to the fact that in these models, the neuronal cells do not undergo differentiation and maturation processes despite the formation of neurites from the neuronal bodies. These processes are essential because neuronal maturation involves the activation and modification of signaling components that interact with CB1 and consequently affect the spatial distribution and concentration of metabolic enzymes and substrates associated with eCBs [44,66]. This spatio-temporal regulation of CB1 distribution also affects CB1 activation in neuronal cultures, which may have opposite effects compared to data from in vivo models [67]. Recent studies highlight that CB1 agonists act as guidance molecules that promote axonal growth when applied homogeneously in cell cultures [68,69,70].

In contrast, when applied discretely allowing the formation of gradients, CB1 agonists induce a collapse of the growth cone followed by a shift in its direction of growth, suggesting that CB1 activation may play a role as repulsive guidance molecules [38,43]. Loss-of-function assays or pharmacological manipulations during fetal development have shown that the absence or inactivation of CB1 leads to abnormalities in the formation of neuronal circuits, particularly during the process of attachment of one axon to another, inhibiting the conformation of axon groups (called fascicles) [43,44,61,63,71].

The mechanisms responsible for CB1 activation leading to changes in growth cone architecture are not yet fully understood. However, it has been suggested that CB1 may modulate the vesicular recycling of receptors for other guidance molecules after its activation, as has been reported for the receptor of the molecule netrin, whose presence in the plasma membrane decreases after CB1 activation in cortical neurons [61]. When CB1 is activated, it can also transactivate other receptors, particularly tyrosine kinase (TK) receptors, such as fibroblast growth factor (FGF) receptors, which are now known to be involved in axonal guidance [52,72,73]. In addition, CB1 can also interact with other proteins that modify elements of the cytoskeleton [55,66,67].

Synapse formation (called synaptogenesis) begins with the arrival of axons in the corresponding innervation zone. During synapse formation, the presence of eCBs has been detected, suggesting that these molecules play a modulatory role in this process [38,44,74]. Endocannabinoids can modulate the excitability of the developing postsynaptic terminals and thus determine the formation of synapses at specific sites [75]. In addition, during synaptogenesis, the rate of degradation and synthesis of eCBs is regulated, which is due to the spatio-temporal restricted expression of enzymes, such as MGL, that shift their expression to growth cones located near postsynaptic neurons [44]. After synapse formation, MAGL is overexpressed at these presynaptic sites, whereas MGL is restricted to dendritic compartments in the rest of the cell body [44]. These studies strongly suggest that eCB signaling is a remarkable modulatory player that enables precise axon direction and triggers growth cone responses during circuit formation in neurodevelopment.

## 4. Clinical Evidence of the Effect of Cannabinoid Consumption During Pregnancy

The consumption of cannabinoids during pregnancy is a topic that has garnered considerable attention within the scientific community. As the legalization of cannabis becomes widespread, there is an increasing need to understand its extensive impacts on both maternal and fetal health. A multitude of significant studies has emerged, providing a detailed examination of the potential risks associated with cannabinoid exposure during pregnancy. These investigations highlight the intricate relationship between prenatal cannabinoid use and developmental outcomes, emphasizing the critical need for continued research in this evolving area of study.

THC and CBD, once absorbed into the maternal bloodstream, are capable of penetrating the placenta and entering the fetal circulatory system, raising significant concerns about the direct exposure of the fetus to these compounds [76]. This penetration may lead to developmental and neurological consequences that are not yet fully delineated. Interestingly, the human placenta expresses cannabinoid receptors that regulate serotonin transporter activity [77,78]. Alterations in these receptors have been associated to increased serotonin levels, which can be transferred to the fetal brain.

### 4.1. Effect of Cannabinoids in the Development of Peripheral Tissues

There is evidence of the association between cannabis use during pregnancy and an increased incidence of anemia among mothers that correlates cannabis consumption with reduced hemoglobin levels, which could precipitate maternal anemia, leading to fatigue, weakness, and severe health complications [79]. The repercussions of maternal anemia could extend to the fetus, potentially hindering development due to inadequate oxygen delivery. Furthermore, results from a systematic review underscored the link between prenatal cannabis exposure and adverse birth outcomes, including preterm delivery. This meta-analysis suggests that cannabis users have a higher propensity for premature births, which are associated with numerous neonatal complications such as underdeveloped lungs and elevated infection risks [80].

Even though cannabinoids bear the potential to mitigate symptoms of hyperemesis gravidarum, one study highlighted a significant gap in the safety data, suggesting that the potential developmental and behavioral risks to the fetus might surpass the benefits of such treatment [81].

### 4.2. Effect of Cannabinoids in the Development of the Nervous System

Cannabinoid exposure on fetal brain development causes permanent alterations in neurotransmitter systems in animal models exposed to THC during gestation [82]. These alterations, particularly in regions of the brain associated with memory and learning, suggest that THC could disrupt normal brain development, potentially leading to sustained cognitive and behavioral impairments. Furthermore, it has been described that prenatal cannabinoid exposure is associated with alterations in the function and expression of CB1, which regulates neurotransmitters such as GABA, glutamate, and acetylcholine that are involved in memory and learning [83]. Moreover, cannabis induces alteration in serotonin signaling and promotes dysregulation in the hypothalamus–pituitary axis, affecting the amygdala and oxytocin levels, a hormone associated to behavioral alterations [78,84]. Additionally, chronic exposure to exogenous cannabinoids in animal models alters long-term synaptic plasticity in the hippocampus through the reduced expression and function of glutamate receptors (GluR1 and GluR2) and CREB phosphorylation [83,85].

In humans, there are consistent links between prenatal THC exposure and challenges in executive functioning and attention in children. These challenges could persist into adulthood, adversely affecting educational achievements and social dynamics [86,87]. Additionally, there are molecular alterations in the fetal brain following marijuana exposure, such as a reduction in the expression of D2 and opioid receptors. Such changes could potentially impact emotional regulation and predispose individuals to substance abuse later in life [6,88]. Furthermore, there are structural changes in the brains of fetuses exposed to marijuana, such as increased cortical thickness, which could correlate with the neurodevelopmental and cognitive issues observed in affected children [5].

There is robust documentation about the behavioral and psychological impacts of prenatal cannabis exposure which collectively highlight an elevated risk of attention deficits, sleep disturbances, and psychiatric disorders such as psychosis [89,90,91,92]. Interestingly, compelling evidence links prenatal cannabis exposure to low birth weight and preterm birth, conditions that carry a high risk of subsequent health complications, including respiratory and neurological disorders [93]. Additionally, clinical data suggest a potential increase in the risk of Sudden Infant Death Syndrome (SIDS) associated with prenatal cannabis exposure, pointing to a complex interplay between prenatal exposure and postnatal environmental factors [94].

On the other hand, there is evidence that raises concerns about the teratogenic effects of cannabinoids, noting an association with congenital heart defects, indicating potential functional and anatomical impacts on the developing fetus [95].

Collectively, these studies create a robust argument for the cautious consideration of cannabis use during pregnancy, given the array of potential adverse effects on both the mother and the developing fetus. Notwithstanding, the direct effect of cannabinoid consumption during human embryonic development will always be a limitation in clinical studies. This issue demands study models to elucidate the impact of cannabinoids on human development.

## 5. Bidimensional and Tridimensional Neuronal Cultures from Human Pluripotent Stem Cells and Induced Pluripotent Stem Cells for Neurodevelopmental Studies

Animal models have played a crucial role in expanding our understanding of how cannabinoids influence neural development. However, they are accompanied by several significant limitations. Key differences in the timing of brain development, variations in the components of the endocannabinoid system, and discrepancies in receptor distribution between humans and animals often constrain the relevance of these findings to human biology [41]. For instance, rodent models are unable to fully replicate the complexity of human neurodevelopment or the unique features of the human endocannabinoid system, making extrapolation of results to humans challenging [96,97]. Additionally, ethical considerations and the absence of a human-specific biological context further limit their applicability for studying prenatal cannabinoid exposure [98].

Importantly, clinical studies addressing prenatal cannabis exposure also face considerable methodological obstacles that can compromise their reliability. A prominent issue is the reliance on self-reported cannabis use, which is frequently affected by recall bias and underreporting due to the stigma surrounding substance use during pregnancy [99]. Moreover, the ability to control confounding variables presents another challenge. Factors such as the concurrent use of tobacco or alcohol, socioeconomic conditions, and pre-existing health issues often interfere with isolating the specific impact of cannabis [80]. Furthermore, variability in study designs, including differences in sample sizes, exposure assessments, and measured outcomes, creates inconsistencies that hinder the comparability and synthesis of results [80].

To overcome these challenges, human-based models, including bidimensional and tridimensional neuronal cultures derived from pluripotent stem cells (PSCs), offer a more accurate alternative. These models enable researchers to study human neurodevelopment and the effects of cannabinoids in a controlled environment that closely mimics human biology [100]. By providing a framework that accounts for the intricacies of human neural development and differentiation, these models offer a powerful tool to address the limitations inherent in animal and clinical studies.

### 5.1. ESCs and iPSCs for Neurodevelopmental Studies

ESCs derive from the inner mass of blastocysts and can give rise to virtually every cellular type of the three embryonic germ layers [101]. Diverse protocols have enabled the derivation of specific types of neurons from PSCs, trying to recapitulate the cellular and structural complexity of the developing human brain and specific brain regions. The establishment of these protocols facilitates functional studies of cerebral development, disease modeling, and drug discovery [102,103,104]. PSCs can generate neurons, employing specific differentiation protocols, in bidimensional cultures or tridimensional cultures as brain organoids, which can recapitulate several aspects of human neurodevelopment; this property allows the study of how genetic and environmental variables affect human neurodevelopment in vitro [105,106]. Although promising, the use of ESCs to study human neurodevelopment and neuronal function is limited by ethical issues regarding their use and obtention due to their embryonic origins. Cell reprogramming is a technology that shifts the phenotype of terminally differentiated somatic cells to an undifferentiated phenotype similar to that of ESCs. The development of technologies to generate hESCs raised the possibility of producing large numbers of defined classes of neurons for research and transplantation. More recently, the development of methods to reprogram adult somatic cells, including fibroblasts, to PSCs, referred to as iPSCs, has made it possible to generate patient-specific iPSCs. The reprogramming procedure was reported for the first time by Takahashi and Yamanaka in 2006 and is mainly performed by the ectopic expression of four pluripotency transcription factors, Oct4, Sox2, c-Myc, and Klf4, in somatic mouse fibroblast cells. A year later, the same authors demonstrate the generation of iPSCs cells from adult human dermal fibroblasts with the same four factors [107,108]. This procedure is mainly performed by the ectopic expression of pluripotency transcription factors. The cells reprogrammed to an undifferentiated phenotype are iPSCs that share many properties with ESCs, such as self-renewal, the capacity to differentiate into every type of cell of the three germinal layers, including neurons, and the property to generate brain organoids [109,110]. 

### 5.2. Unpatterned and Patterned 3D Cell Cultures

In recent years, 3D neuronal cultures derived from ESCs and iPSCs have provided new opportunities for investigating human brain development and diseases. In 2013, Lancaster, Knoblich and coworkers established a 3D culture system to generate brain tissue from human PSCs, termed cerebral organoids, that develop various discrete, although interdependent, brain regions, which closely mimics the endogenous developmental program [105,111]. These culture systems offer a more physiologically relevant environment than traditional 2D cultures, enabling more accurate modeling of the complex architecture and functionality of the human brain and overcoming the limitations found in 2D cell cultures and experimental animal models.

Cerebral organoids are 3D cultures that model organogenesis and provide a new platform to investigate human brain development. Most studies have focused on generating minimally patterned brain organoids, providing a basic medium, that developed into structures that contain multiple brain regions, including the cerebral cortex, choroid plexus, ganglionic eminence hippocampus, and retina, which exploit the intrinsic self-organizing properties of PSC-derived aggregates. However, these unguided approaches do not allow a systematic and reproducible assessment, as brain regions appear to be distributed randomly [105,111]. Recent studies with patterned organoids direct regional fate specification through patterning factors such as morphogens and small molecules. For instance, human cortical organoids recapitulate the complex 3D structure and physiological function of human brain: they have progenitor zone organization, neurogenesis, gene expression, and, notably, a distinct human-specific outer radial glia cell layer [112,113]. Additionally, patterned brain region-specific organoids have been generated for the midbrain [114], hippocampus [115], cerebellum [116], hypothalamus [117,118], and spinal cord [119]. These organoids recapitulate key features of specific brain regions, have minimal heterogeneity, and provide a platform for modeling human brain development and disease and for compound testing.

### 5.3. Organoids for Neurodevelopmental Studies

Three-dimensional neuronal cultures, often called brain organoids, model the structural organization of the cellular composition and connectivity of the brain. They are generated by differentiating ESCs or iPSCs into neural progenitor cells, which then self-organize into spheroids or organoids exhibiting key brain tissue properties [120]. These properties include the formation of distinct neuronal layers, the presence of various neural cell types, the development of synaptic connections, and the transcriptional and epigenetic signature of a developing human brain and recapitulate crucial molecular and cellular steps of brain development [121,122,123]. Knowledge of normal organ developmental pathways guides the formation of these structures, and the incorporation of single-cell sequencing, genome editing, and optogenetics could improve the application of brain organoids.

One of the main advantages of using PSCs for generating 3D neuronal cultures is their ability to differentiate into any cell type, providing a limitless source of human neurons and glial cells that can be used to model development and disease. Particularly, iPSCs are derived from adult somatic cells reprogrammed into a pluripotent state that allow the generation of patient-specific neural models. This property makes possible to study the genetic basis and cellular mechanisms underlying neurodevelopmental disorders in a patient-specific context, building the way for personalized medicine approaches.

Recent studies have demonstrated the utility of 3D neuronal cultures in modeling neurodevelopmental diseases such as microcephaly, autism spectrum disorder (ASD), schizophrenia, and epilepsy [123,124]. For instance, brain organoids derived from iPSCs of individuals with ASD have revealed aberrant neurodevelopmental processes, such as a significant decrease in cell cycle length, altered neuronal migration, abnormal synapse formation, and an increase in the number of inhibitory synapses compared to the control organoids [122]. Similarly, organoids generated from iPSCs of patients with epilepsy have shown dysregulated neuronal network activity, providing insights into the disease’s pathophysiology and potential therapeutic targets [124].

Altogether, this information indicates that 3D neuronal cultures from ESCs and iPSCs represent a transformative tool in neurodevelopmental research. They provide a robust platform for investigating the cellular and molecular mechanisms of brain development and disorders, offering promising avenues for therapeutic discovery and personalized medicine. Continuing innovation in this field holds the potential to unravel the complexities of the human brain and make significant progress in our understanding of neurodevelopmental diseases.

Furthermore, 3D neuronal cultures offer a valuable platform for drug screening and toxicity testing [125]. The more accurate recapitulation of the brain’s microenvironment allows for the assessment of drug efficacy and safety in a context that closely mimics human physiology. This feature is particularly important for neurodevelopmental disorders, where animal models are limited in revealing some of the most fundamental aspects of development, genetics, pathology, and disease mechanisms that are unique to humans, such as neocortical development, including cortical expansion, a protracted time of development, and genetics [126,127].

### 5.4. Future Directions in 3D Cell Cultures

Despite these advancements, several challenges remain in 3D neuronal cultures. The reproducibility of organoid generation, the heterogeneity of the cultures, and the lack of vascularization require further improvement [120]. Advances in bioengineering and the integration of microfluidic systems may help to address these issues, improving the fidelity and functionality of 3D neuronal models.

Altogether, this information indicates that 3D neuronal cultures from PSCs represent a transformative tool in neurodevelopmental research [121]. They provide a robust platform for investigating the cellular and molecular mechanisms of brain development and disorders, offering promising avenues for therapeutic discovery and personalized medicine. Continuing innovation in this field holds the potential to unravel the complexities of the human brain and make significant progress in our understanding of neurodevelopmental diseases.

## 6. Evidence of the Effect of Cannabinoids on Human Bidimensional and Tridimensional Neuronal Cultures

### 6.1. Effect of Cannabinoids in PSCs

The information regarding the impact of cannabinoid consumption on the early stages of human embryogenesis is scarce. Evidence from mouse pluripotent stem cells (mPSCs) indicates that cannabinoid receptors display a modest expression in this type of cells. This result supports the finding that THC does not modify the migration or proliferation of mPSCs; however, their blockade impairs their integrity, suggesting a role of CB receptors on mPSCs even at low expression levels [128]. When cultured as embryoid bodies, mPSCs also display a modest expression of CB receptors, although their activation by WIN agonist promotes a slight increase in size of embryoid bodies and a reduction on cell death; meanwhile, it does not increase proliferation [129]. These results suggest that it is likely that human PSCs also display low levels of cannabinoid receptors; nonetheless, whether human PSCs express cannabinoid receptors that modulate their cell biology needs to be addressed. On the other hand, recent evidence on human ESC and iPSC platforms demonstrates the active role of cannabinoid receptors on human neurodevelopment (Table 1).

### 6.2. Effect of Cannabinoids in Neuronal Differentiation of Human Neurons and Brain Organoids

Novel information demonstrates that THC impairs the formation of human cortical organoids derived from PSCs. THC decreases organoid maturation, neurite outgrowth, and the expression of CB1 receptors. This information is in line with further results that show a decrease in the neuronal activity of cortical organoids. Conversely, THC increases the proliferation of neural progenitors [130]. This information suggests that THC impairs the maturation of the human cortex by promoting an undifferentiated state.

Further findings indicate that cortical spheroid cultures derived from human iPSCs express the enzymes to synthetize endocannabinoids. Cortical spheroid cultures also have neurons that express functional CB1 receptors to respond to cannabinoids or their agonists. Interestingly, antagonists of CB1 receptors increase their firing rate and increase or decrease their bursting frequency in a concentration-dependent manner [131]. This evidence supports the findings on cortical organoids as it highlights that the exogenous exposure to an additional source of cannabinoids can compromise the correct neuronal function and suggests that endocannabinoid signaling is essential in human neurodevelopment to modulate neuronal activity.

Interestingly, studies on brain organoids derived from human iPSCs further support that cortical neurons express CB1 receptors. The exposure of brain organoids to cannabinoid agonists or THC increases the number of deep cortical layer neurons. Conversely, activating cannabinoid receptors with these molecules decreases the SATB2-positive cortical neurons in this study model. This information is consistent with the results obtained from brain organoids of mouse embryonic stem cells, indicating that not only does the activation of the CB1 receptor by exogenous cannabinoids impair cortical layer formation but also that this process is conserved among species due to the similar responses between mouse and human cells [132].

Studies on bidimensional cultures of human cortical neurons derived from iPSCs demonstrate that cannabinoids and their agonists modulate neuronal activity. In line with the previous evidence, the CB receptor agonist Win 55,212-2 and the endocannabinoid 2-AG decreased neuronal calcium signaling, suggesting their role as negative modulators of neuronal activity [100]. Remarkably, this information further underscores the modulatory activity of cannabinoids on neuronal function, indicating that these compounds decrease neuronal activity regardless of whether they are studied on bidimensional or tridimensional cultures. These consistent findings provide robust evidence of the essential role of cannabinoids on neural development.

Since glutamatergic neurons comprise a large population of cells in the cortex, it is likely that they are susceptible to the exposure of endogenous and exogenous cannabinoids. In line with this evidence, human glutamatergic neurons derived from iPSCs are susceptible to THC and endocannabinoids since these molecules decrease their neurite outgrowth. Furthermore, this event correlates with a decrease of Erk1/2 phosphorylation and the blockade of cannabinoid signaling through CB1 inhibits the effect of cannabinoids on neurite length, thus suggesting that neurite growth impairment is promoted by a decrease of Erk1/2 signaling [133]. Therefore, the impairments of neuronal activity may be partially mediated by alterations of the structural plasticity of human glutamatergic neurons that reside within the developing cortex. This suggestion is in line with previous findings describing a decrease of neuronal activity on cortical spheroid cultures, cortical organoids, and bidimensional cultures of cortical neurons, which together strengthen the notion that cannabinoids decrease neuronal activity through impairments of structural plasticity.

On the other hand, a study with phytocannabinoids, cannabidiol and THC, and synthetic analogs demonstrated that these compounds accelerated the neuronal and glial differentiation of bidimensional cultures of neurons derived from human iPSCs. The phytocannabinoids and their analogs decreased calcium signaling in neurons, which is consistent with the evidence that demonstrates that exogenous cannabinoids alter neuronal activity. Remarkably, only the synthetic analog THJ-018 induced a modest increase in cell death in neuronal cultures derived from human iPSCs [134]. This information suggests that while cannabinoids and their analogs promote cellular impairments, including altered neuronal activity or increased rate of differentiation, their potential to induce neuronal death is minimal in human neurons. The discrepancies in the results regarding neuronal differentiation with the previous studies may be attributed to the different protocols of neuronal differentiation, since this research group did not use a protocol to yield a culture enriched with cortical neurons. However, after taking all this information together, there is robust evidence indicating that exogenous cannabinoids impair neuronal activity in cortical neurons by decreasing it, presumably through morphological alterations of glutamatergic neurons; moreover, both in vivo animal and in vitro organoid studies suggest that the effect of cannabinoids on neurogenesis has a significant negative impact on glutamatergic cortical neurons directly associated with cognitive and behavioral development (Figure 3).

In contrast to cortical neurons, cannabinoid exposure does not appear to impair the differentiation of human dopaminergic neurons derived from human neuronal precursors [135]. Furthermore, cannabinoids promote neither neuronal death nor epigenetic changes. However, cannabinoids modulate the electrophysiological properties of human dopaminergic neurons by decreasing the amplitude of their action potentials and their post-synaptic currents, thus strongly suggesting that cannabinoids decrease neuronal activity regardless of the neuronal phenotype, and further studies analyzing other types or neurons will shed light on the effects of cannabinoids on the developing nervous system of humans.

Altogether, the information obtained from neurons and brain organoids derived from human iPSCs and exposed to cannabinoids points to a decrease of the neuronal activity caused by these molecules in different types of neurons. This effect may be partially attributed to a decrease on the differentiation process to a neuronal phenotype (Table 1; Figure 3). Further knowledge using human 2D and 3D cultures regarding to the role of cannabinoids on biological processes such as synaptogenesis and axonal growth remains to be unveiled.

While ESC and iPSC models have provided transformative insights into human neurodevelopment and the effects of cannabinoids, they come with inherent limitations that must be carefully considered. One of the most pressing challenges is the reproducibility of findings across different experiments and laboratories [136]. Differences in protocols of differentiation, culture conditions, and even variations between ESC or iPSC batches can introduce significant variability in results [137]. For example, inconsistencies in growth factors, timing, or substrate composition can alter the cellular outcomes of organoid development, making it challenging to standardize methodologies [138]. Such variability not only complicates the interpretation of findings but also limits the broader applicability of these models, particularly in comparative studies [139,140].

Adding to this complexity is the heterogeneity within organoid cultures. Brain organoids often lack uniformity in terms of cellular composition, spatial organization, and developmental progression [141]. This heterogeneity arises from their self-organizing nature, which, while useful for recapitulating some aspects of brain development, falls short of replicating the tightly regulated processes observed in vivo [120]. For instance, differences in the proportion of excitatory versus inhibitory neurons or the presence of immature cell types can lead to inconsistent functional outputs, limiting their utility for specific research questions [142]. Furthermore, their inability to mimic intricate cell–cell interactions or regionalized brain structures highlights the gap between organoid models and the in vivo brain environment [143,144].

Another critical limitation is the absence of vascularization. A functional vascular system is essential for delivering nutrients and oxygen to developing tissues, especially in regions of the brain that are metabolically demanding [145]. Without vascularization, organoid growth and maturation are restricted and regions farthest from the nutrient supply often suffer from hypoxia-induced cell death [146]. This issue not only impacts the overall viability of the organoid but also restricts its ability to replicate the metabolic dynamics of a developing brain. Recent bioengineering approaches, such as the incorporation of endothelial cells or the use of microfluidic systems, have shown promise in introducing vascular-like structures into organoids. However, these methods remain experimental and are not yet widely implemented in standard protocols [146,147].

## 7. Conclusions

Cannabinoids play crucial roles in neurodevelopment among different species, including humans. The accumulating evidence strongly suggests a tight regulation of these molecules during neurodevelopment that, when modified, could bring serious consequences for the human brain’s integrity and function. Although unraveling the mechanisms through which exogenous cannabinoids affect the developing human brain represents a challenging task, new technologies such as iPSC and ESC platforms have arisen to address these issues and have demonstrated that exogenous cannabinoids alter neuronal properties and neuronal differentiation. Therefore, further research using these models will be essential to understand human neurodevelopment in the presence of exogenous cannabinoids.

## 8. Future Directions

The use of ESCs and iPSCs has revolutionized the study of neurodevelopmental problems in humans since these platforms allow the evaluation of the effects of multiple molecules in the process of neuronal differentiation. It is expected that PSCs and iPSCs will be further exploited to gather more information about the impact of exogenous cannabinoids on human neurodevelopment and further experiments using iPSCs could also shed light on the way these molecules alter the formation of the nervous system according to the sex, ethnicity, and/or genetic background of individuals. Nevertheless, there will also be a need for improving the reproducibility of results and attending to the bioethical concerns of these study models. Addressing these perspectives will pave the way to a more precise understanding of the real impact of cannabinoid consumption during pregnancy that will assist in the elaboration of more robust politics of health care regarding this issue.

## Figures and Tables

**Figure 1 cells-14-00070-f001:**
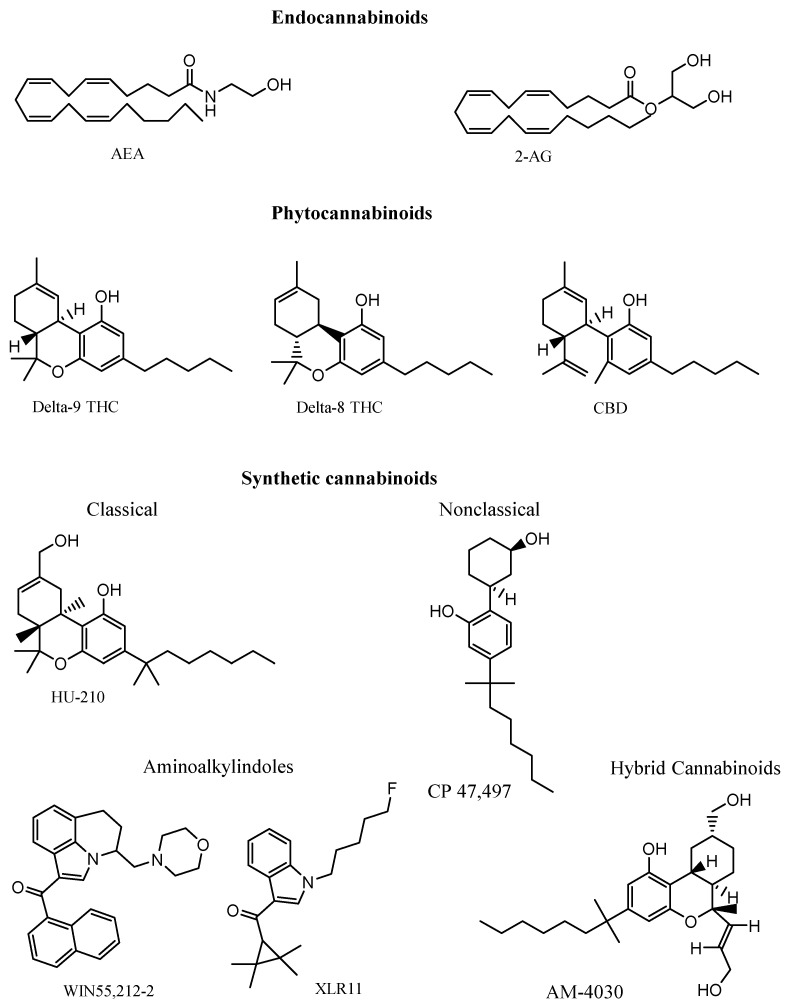
Endocannabinoids are natural agonists produced by the enzymes of neurons; phytocannabinoids are produced exclusively in the secondary metabolism of the Cannabis sativa plant; and synthetic cannabinoids, which are new molecules with a very wide structural diversity, are currently prohibited for synthesis, use, and distribution by the general public around the world. Abbreviations: AEA, anandamide; 2-AG, 2-arachidonoylglycerol; CBD, cannabidiol; THC, tetrahydrocannabinol.

**Figure 2 cells-14-00070-f002:**
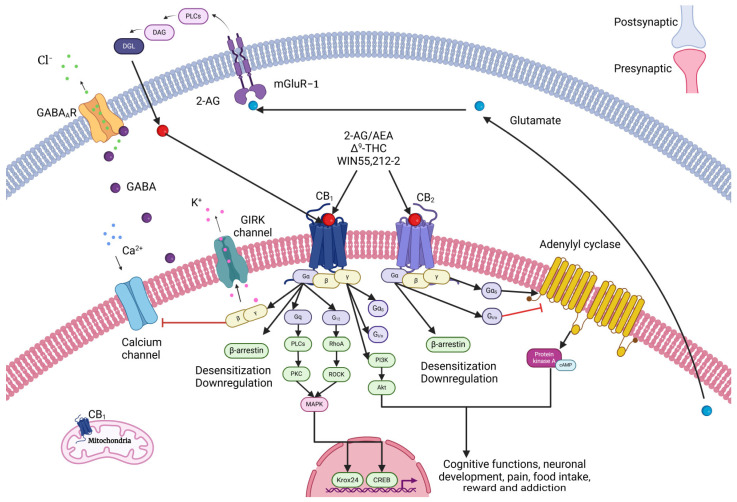
Signal transduction of CB1 and CB2. The postsynaptic neuron synthetizes and releases eCBs on the presynaptic terminal, where they bind to the cannabinoid receptors present in the presynaptic neuron. Once an eCB binds its receptor, there is a presynaptic inhibition of the Ca^2+^ flow, promoted by the inhibition of the CaV-type calcium channels and the activation of the G Protein-Coupled, Inward-Rectifying K^+^ (GIRK) channels. Additionally, the activation of the GIRK channels reduces calcium flow due to an increase in K^+^ conductance, producing hyperpolarization. This Ca^2+^ flow is responsible for a balance between the processes of Ca^2+^-dependent activation, such as muscle contraction and the release of neurotransmitters. Created with Biorender.com. Modified from Fletcher-Jones et al. (2020); Leo and Abood (2021); Gasperi et al. (2023) [23,32,33].

**Figure 3 cells-14-00070-f003:**
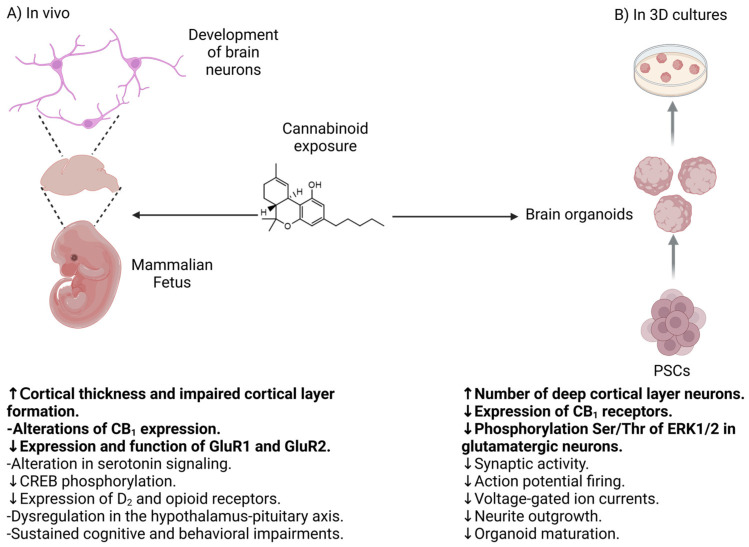
Similarities in the effect of Δ9-THC and other cannabinoids on neurodevelopment using in vivo models and organoids (sentences in bold), as well as other findings identified in both models. (**A**) In vivo model and (**B**) in vitro model with organoids. Created with Biorender.com.

**Table 1 cells-14-00070-t001:** Cannabinoid effects on neuronal cells derived from human iPSCs.

Platform Study	Drug	Biological Effects	References
hiPSC-derived neuron/astrocyte cultures	Win-2 Δ9-THC 2-AG Carbachol	hiPSC-derived neuronal cultures displayed a glutamatergic signaling network and sensitivity to synthetic cannabinoids and endocannabinoids. Additionally, these cells produced endocannabinoids, like 2-AG, in response to carbachol in a Ca^2+^-independent manner, while Δ9-THC acted as a partial agonist inhibiting synaptic activity.	[100]
Human cerebral organoids from ESCs	Δ9-THC	Prolonged exposure to Δ9-THC showed brain organoids with altered neurite outgrowth, reduced neuronal maturation, and down-regulation of CB1 expression.	[130]
Cortical spheroid model of neurodevelopment	CB1-selective antagonist SR141716A	CB1 receptors regulated the synaptic strength and the balance of inhibitory and excitatory signaling. Acute SR141716A treatment contributed to demonstrate the role of the endocannabinoid system in regulating neuronal connections and synaptic activity.	[131]
hiPSC-derived cerebral organoids	HU-210, CB1 agonist THC	CB1 receptor participated in neuronal differentiation of deep layer neurons. THC and CB1 agonist HU-210 promoted the expansion of BCL11B+ neurons, meanwhile reducing the number of SATB2+ upper layer neurons.	[132]
Cortical neurons derived from human hiPSCs	2-AG Δ9-THC SR 141716A, CB1 Selective inverse agonist	Cortical neurons derived from hiPSCs expressed CB1 and responded to cannabinoids; 2AG and Δ9-THC reduced neurite outgrowth and phosphorylation of serine/threonine kinase extracellular signal-regulated protein kinases (ERK1/2), while Δ9-THC reduced phosphorylation of Akt. These effects could be blocked by a CB1 receptor antagonist.	[133]
Neuronal cells derived from human induced pluripotent stem cells (hiPSCs)	Cannabidiol (CBD) Δ9-THC THJ-018 and EG-018, synthetic cannabinoids (SCs)	These compounds promoted alterations in neuronal development; CBD is neurotoxic, reducing cellular density in cultures of neuronal progenitor and differentiated neurons. SCs and Δ9-THC induced premature glial and neuronal differentiation and abnormal function of voltage-gated calcium channels in neurons.	[134]
Human cord blood-derived induced pluripotent stem cell (hCBiPSC)-derived small molecule neural precursor cells (smNPSc)	Anandamide (AEA), endogenous cannabinoid, CB1 receptor agonist Δ9-THC, exogenous cannabinoid, CB1 receptor agonist	High concentrations (10 µM) of AEA or THC during human neurogenesis reduced synaptic activity, action potential firing, and voltage-gated ion currents, affecting neuronal functionality, while low concentrations (1 µM) of AEA increased synaptic activity in neurons.	[135]

## Data Availability

Data is contained within the article.

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
