# Peer review of "Modeling the Effect of Cannabinoid Exposure During Human Neurodevelopment Using Bidimensional and Tridimensional Cultures"

_cells, 2025, doi:10.3390/cells14020070_

Round 1

Reviewer 1 Report

Comments and Suggestions for Authors

This paper provides a interesting review of the complex role of cannabinoids in human neurodevelopment analised through the use of 2D and 3D cultures of human pluripotent stem cells (PSCs) and induced pluripotent stem cells (iPSCs). It builds a compelling argument for the cautious use of cannabis during pregnancy.

The authors provide a detailed explanation of the different types of cannabinoids, their receptors, and their signalling pathways in the mature nervous system. This summary is important for understanding the subsequent discussion on the impact of cannabinoids during neurodevelopment.

Overall, the paper is well-written and effectively argues for the importance of studying the effects of cannabinoids on human neurodevelopment. The authors' use of in vitro models, is particularly commendable, as it addresses the limitations of clinical studies and offers promising avenues for future research.

Before considering publication I would suggest to improve this text in order to increase its impact:

1)    The paper's introduction spends considerable time discussing theproperties of cannabinoids in the mature nervous system. While some background is necessary, this extensive detail feels disproportionate, particularly when the focus of the paper is on neurodevelopment. The excessive focus on mature systems detracts from the paper's core objective. For instance, detailed descriptions of CB1 and CB2 distribution and function in mature brains, while interesting, are not strictly relevant to understanding their roles in neurodevelopment. A more concise introduction focusing on cannabinoid signalling relevant to developmental processes might be been more effective.

2)    Although the authors acknowledge the limitations of clinical studies, they fail to critically evaluate the quality and consistency of the evidence presented. The paper would benefit from a more in-depth analysis of the methodological challenges inherent in these studies, such as the reliance on self-reported cannabis use, the difficulty in controlling for confounding factors, and the variability in study designs. A more nuanced discussion of these limitations would strengthen the argument for using in vitro models.

3)    The paper presents PSCs and iPSCs as a panacea for the limitations of clinical studies, without adequately addressing the challenges and limitations of these models. For example, the paper does not discuss the issues of reproducibility, heterogeneity, and lack of vascularization in brain organoids, which are crucial considerations for interpreting findings. A more balanced discussion that acknowledges these limitations would enhance the paper's scientific rigour.

4)    The paper presents a list of findings from in vitro studies without synthesizing sufficiently well the information or critically evaluating the studies limitations. Simply listing results from various studies does not provide a coherent or insightful understanding of the complex interplay between cannabinoids and neurodevelopment. The paper would benefit from a more analytical approach, identifying common themes and discrepancies between studies, and discussing the implications of these findings in a broader context. For instance, the paper could compare the effects of different cannabinoids (THC, CBD, 2-AG) on various neuronal subtypes and developmental processes across different studies, highlighting consistent patterns and knowledge gaps.

5)    Few more figures will benefit the readability of the paper.

Author Response

REVIEWER 1

Comment 1:    The paper's introduction spends considerable time discussing the properties of cannabinoids in the mature nervous system. While some background is necessary, this extensive detail feels disproportionate, particularly when the focus of the paper is on neurodevelopment. The excessive focus on mature systems detracts from the paper's core objective. For instance, detailed descriptions of CB1 and CB2 distribution and function in mature brains, while interesting, are not strictly relevant to understanding their roles in neurodevelopment. A more concise introduction focusing on cannabinoid signaling relevant to developmental processes might be been more effective.

Response 1: We appreciate your important comments on this paper. We decided to explain the signaling pathway as it is known in the mature nervous system at the initial sections because we consider that many of the reviews have incomplete information that we consider essential to better understand the role of the endocannabinoid system in neurodevelopment

Comment 2:    Although the authors acknowledge the limitations of clinical studies, they fail to critically evaluate the quality and consistency of the evidence presented. The paper would benefit from a more in-depth analysis of the methodological challenges inherent in these studies, such as the reliance on self-reported cannabis use, the difficulty in controlling for confounding factors, and the variability in study designs. A more nuanced discussion of these limitations would strengthen the argument for using in vitro models.

Response 2: We thank the reviewer comment. This information will substantially increase the quality of the manuscript. We added the requested material between lines 438 and 447 highlighted in yellow and written as follows:

“Importantly, clinical studies addressing prenatal cannabis exposure also face considerable methodological obstacles that can compromise their reliability. A prominent issue is the reliance on self-reported cannabis use, which is frequently affected by recall bias and underreporting due to the stigma surrounding substance use during pregnancy [101]. Moreover, the ability to control for confounding variables presents another challenge. Factors such as concurrent use of tobacco or alcohol, socioeconomic conditions, and pre-existing health issues often interfere with isolating the specific impact of cannabis [81]. Furthermore, variability in study designs, including differences in sample sizes, exposure assessments, and measured outcomes, creates inconsistencies that hinder the comparability and synthesis of results [81]”.

Comment 3:    The paper presents PSCs and iPSCs as a panacea for the limitations of clinical studies, without adequately addressing the challenges and limitations of these models. For example, the paper does not discuss the issues of reproducibility, heterogeneity, and lack of vascularization in brain organoids, which are crucial considerations for interpreting findings. A more balanced discussion that acknowledges these limitations would enhance the paper's scientific rigour.

Response 3: We thank the valuable opinion of reviewer. To provide a more balanced discussion addressing the limitations of PSC and iPSCs we added the following information between lines 682 and 711 highlighted in yellow:

“While ESCs and iPSCs models have provided transformative insights into human neurodevelopment and the effects of cannabinoids, they come with inherent limitations that must be carefully considered. One of the most pressing challenges is the re-producibility of findings across different experiments and laboratories [140]. Differences in protocols of differentiation, culture conditions, and even variations between ESCs or iPSC batches can introduce significant variability in results  [141]. For example, inconsistencies in growth factors, timing, or substrate composition can alter the cellular outcomes of or-ganoid development, making it challenging to standardize methodologies [142]. Such variability not only complicates the interpretation of findings but also limits the broader applicability of these models, particularly in comparative studies [143,144].

Adding to this complexity is the heterogeneity within organoid cultures. Brain organoids often lack uniformity in terms of cellular composition, spatial organization, and developmental progression [145]. This heterogeneity arises from their self-organizing nature, which, while useful for recapitulating some aspects of brain development, falls short of replicating the tightly regulated processes observed in vivo [146]. For instance, differences in the proportion of excitatory versus inhibitory neurons or the presence of immature cell types can lead to inconsistent functional outputs, limiting their utility for specific research questions [147]. Furthermore, their inability to mimic intricate cell-cell interactions or regionalized brain structures highlights the gap between organoid models and the in vivo brain environment [148,149].

Another critical limitation is the absence of vascularization. A functional vascular system is essential for delivering nutrients and oxygen to developing tissues, especially in regions of the brain that are metabolically demanding [150]. Without vascularization, organoid growth and maturation are restricted, and regions farthest from the nutrient supply often suffer from hypoxia-induced cell death [151]. This issue not only impacts the overall viability of the organoid but also restricts its ability to replicate the metabolic dynamics of a developing brain. Recent bioengineering approaches, such as the incorporation of endothelial cells or the use of microfluidic systems, have shown promise in introducing vascular-like structures into organoids. However, these methods remain experimental and are not yet widely implemented in standard protocols [151,152]”.

Please notice that in some sentences we changed the term pluripotent stem cells (PSCs) to embryonic stem cells (ESCs) to differentiate ESCs from iPSCs in a more accurate manner.

Comment 4:    The paper presents a list of findings from in vitro studies without synthesizing sufficiently well the information or critically evaluating the studies limitations. Simply listing results from various studies does not provide a coherent or insightful understanding of the complex interplay between cannabinoids and neurodevelopment. The paper would benefit from a more analytical approach, identifying common themes and discrepancies between studies, and discussing the implications of these findings in a broader context. For instance, the paper could compare the effects of different cannabinoids (THC, CBD, 2-AG) on various neuronal subtypes and developmental processes across different studies, highlighting consistent patterns and knowledge gaps.

Response 4: In attention to the reviewer comments, we have modified and highlighted the sections where we indicate the common themes and discrepancies between studies of iPSCs derived neurons or organoids exposed to cannabinoids. Additionally we also highlighted the implications of these findings in a broader context (Lines: 605-609, 614-618, 623-627, 635-640, 647-653, 656-659).

Finally, we added a paragraph in which we describe the main findings provided by the studies mentioned in this section which is highlighted in yellow between lines 674-680 and it is written as follows:

“Altogether, the information obtained from neurons and brain organoids derived from human iPSCs and exposed to cannabinoids points out to a decrease of the neuronal activity caused by these molecules in different types of neurons. This effect may be partially attributed to a decrease on the differentiation process to a neuronal phenotype (Table 1; Figure 3). Further knowledge using human bidimensional and tridimensional cultures regarding to the role of cannabinoids on biological processes such as synaptogenesis and axonal growth, remains to be unveiled”.

Comment 5:    Few more figures will benefit the readability of the paper.

Response 5: In attention to this valuable advice, we added a third figure where we show the similarities and advantages of using organoids compared to in vivo neurodevelopmental models. This figure also addresses the concern of the comment 4.

Reviewer 2 Report

Comments and Suggestions for Authors

In the manuscript by Enrique Estudillo et al., the authors provide a comprehensive review of the effects of endocannabinoids used during pregnancy and their impact on neurodevelopment. While the topic is of great relevance and interest, several issues need to be addressed before the manuscript can be considered for publication.

As a general note, it is preferable to cite original research studies rather than secondary reviews when possible. Additionally, the following points should be considered:

  • Reference 1: The foundation of this manuscript hinges on the assertion that cannabinoid use during pregnancy, particularly for treating morning sickness, has increased. It may also be worthwhile to consider other potential reasons for cannabinoid use during pregnancy. For instance, in lines 322–323, the manuscript attributes the need for further knowledge on this topic to cannabis legalization. This statement should be rephrased for clarity and coherence.
  • Presence of CB1 and CB2 receptors: On page 2, lines 45–46, the manuscript states that the presence of endocannabinoid receptors CB1 and CB2 in the developing nervous system strongly suggests that exposure to endocannabinoids could impair development. However, their presence primarily indicates that the nervous system can respond to circulating endocannabinoids. This section would benefit from briefly discussing the physiological roles of endocannabinoids, as extensively covered in chapters 2 and 3.
  • Lines 55–56: The sentence requires rephrasing for accuracy. Pluripotent stem cells (PSCs) do not replicate the development of multiple brain structures. Instead, they model the cell types within these structures. Please clarify this point.
  • Lines 395–396: Please specify that PSCs can generate neurons by applying differentiation protocols.
  • Line 403: Please cite the seminal work of Yamanaka, which is essential to the field of cellular reprogramming.
  • Line 410: Please include foundational studies by Kadoshima and Lancaster, which were instrumental in establishing brain organoid cultures.
  • Lines 408–414: When introducing brain organoids, consider describing the two primary protocols for generating them—minimally patterned and patterned—and highlight what each approach models.
  • Line 415: Please replace "mimic" with "model," as brain organoids serve as models for the cell composition and organization of the human brain.
  • Line 439: Reference 88 (Mariani, 2012) appears outdated for this context. Please include a more recent review on cortical development or insert Reference 89 in this location.
  • Lines 446–447: The claim that animal models fall short in capturing the complexity of human brain development should be rephrased for precision. Indeed they fall short in replicating cell diversity and human genetics.
  • Line 459: A reference is missing. Please ensure this section is appropriately cited.

By addressing these issues, the manuscript will present a more rigorous and coherent analysis of the literature.

Author Response

REVIEWER 2

Comment 1: As a general note, it is preferable to cite original research studies rather than secondary reviews when possible.

Response 1: In attention to the reviewer comment, we have included the original references in the sections where they were needed.

Comment 2: The foundation of this manuscript hinges on the assertion that cannabinoid use during pregnancy, particularly for treating morning sickness, has increased. It may also be worthwhile to consider other potential reasons for cannabinoid use during pregnancy. For instance, in lines 322–323, the manuscript attributes the need for further knowledge on this topic to cannabis legalization. This statement should be rephrased for clarity and coherence.

Response 2: In attention to the reviewer comments, we added other potential reasons for cannabinoids use during pregnancy between lines 39 and 44. This information is highlighted in yellow and written as follows:

“The use of cannabinoids during pregnancy has seen a noticeable increase in recent years, influenced by factors that extend beyond managing morning sickness. Many pregnant women turn to cannabinoids to help alleviate stress, insomnia, or pain. This growing trend may also stem from the widespread perception of cannabis as a natural or relatively safe remedy. Moreover, its greater availability due to legalization in various regions has contributed to its accessibility”

Comment 3: Presence of CB1 and CB2 receptors. On page 2, lines 45–46, the manuscript states that the presence of endocannabinoid receptors CB1 and CB2 in the developing nervous system strongly suggests that exposure to endocannabinoids could impair development. However, their presence primarily indicates that the nervous system can respond to circulating endocannabinoids. This section would benefit from briefly discussing the physiological roles of endocannabinoids, as extensively covered in chapters 2 and 3.

Response 3: We agree with the observation of the reviewer. To address this point, we highlighted that the nervous system can respond to circulating endocannabinoids and added brief information regarding the physiological roles of endocannabinoids. This information is highlighted in yellow between lines 48 and 49.

Comment 4: Lines 55–56. The sentence requires rephrasing for accuracy. Pluripotent stem cells (PSCs) do not replicate the development of multiple brain structures. Instead, they model the cell types within these structures. Please clarify this point.

Response 4: In attention to the reviewer concern, we rephrased and clarified the requested sentence which is highlighted in yellow between lines 62 and 64

Comment 5: Lines 395–396. Please specify that PSCs can generate neurons by applying differentiation protocols.

Response 5: Thank you for pointing this out. We agree with this comment. We added between lines 457 and 461 the following sentence and references: Diverse protocols have enabled the derivation of specific types of neurons from PSC, trying to recapitulate the cellular and structural complexity of the developing human brain and specific brain regions. The establishment of these protocols facilitate functional studies of cerebral development, disease modeling and drug discovery (104-106).

Comment 6. Line 403. Please cite the seminal work of Yamanaka, which is essential to the field of cellular reprogramming.

Response 6: Thank you for pointing this out. We have, accordingly, cited the work of Yamanaka in line 477 to emphasize this point:

“The development of technologies to generate hESCs raised the possibility of producing large numbers of defined classes of neurons for research and transplantation. More recently, the development of methods to reprogram adult somatic cells, including fibroblasts, to pluripotent cells, referred to as induced pluripotent stem cells (iPSCs) has made it possible to generate patient-specific PSCs. The reprogramming procedure was reported for the first time by Takahashi and Yamanaka in 2006 and is mainly performed by the ectopic expression of four pluripotency transcription factors, Oct4, Sox2, c-Myc and Klf4, in somatic, adult fibroblasts, cells. A year later, the same authors demonstrate the generation of iPS cells from adult human dermal fibroblasts with the same four factors (109,110)”

This information is between lines 469 and 477 highlighted in yellow. Please notice that we changed the term pluripotent stem cells (PSCs) to embryonic stem cells (ESCs) to differentiate ESCs from iPSCs in a more accurate manner.

Comment 7: Line 410.Please include foundational studies by Kadoshima and Lancaster, which were instrumental in establishing brain organoid cultures.

Response 7: Thank you for your comment. We agree and have cited the work of Kadoshima and Lancaster in line 495 to emphasize this point:

In 2013, Lancaster and Knoblich established a three-dimensional (3D) culture system to generate brain tissue from human pluripotent stem cells, termed cerebral organoids, that develop various discrete, although interdependent, brain regions, which closely mimics the endogenous developmental program (113,114).

This information is between lines 492 and 495 highlighted in yellow.

Comment 8: Lines 408–414. When introducing brain organoids, consider describing the two primary protocols for generating them—minimally patterned and patterned—and highlight what each approach models.

Response 8: Thank you for your comment. We agree and have described both approaches between lines 500 and 516:

“Cerebral organoids are three-dimensional cultures that model organogenesis and provide a new platform to investigate human brain development. Most studies have focused on generating minimally patterned brain organoids, providing a basic medium, that developed into structures that contain multiple brain regions, including the cerebral cortex, choroid plexus, ganglionic eminence hippocampus and retina, which exploit the intrinsic self-organizing properties of PSC derived aggregates. However, these unguided approaches do not allow a systematic and reproducible assessment, as brain regions appear to be distributed randomly (113,114)

Recent studies with patterned organoids direct regional fate specification through patterning factors such as morphogens and small molecules.

For instance, human cortical organoids recapitulate complex 3D structure and physiological function of human brain: they have progenitor zone organization, neurogenesis, gene expression, and, notably, a distinct human-specific outer radial glia cell layer. (115,116).

Additionally, patterned brain region specific organoids have been generated for midbrain (117), hippocampus (118), cerebellum (119), hypothalamus (120,121) and spinal cord (122). These organoids recapitulate key features of specific brain regions and have minimal heterogeneity and provide a platform for modeling human brain development and disease and for compound testing”.

This information is highlighted in yellow.

Please notice that in some sentences we changed the term pluripotent stem cells (PSCs) to embryonic stem cells (ESCs) to differentiate ESCs from iPSCs in a more accurate manner.

Comment 9: Line 415. Please replace "mimic" with "model," as brain organoids serve as models for the cell composition and organization of the human brain.

Response 9: Thank you for pointing this out. We have, accordingly, replaced de word mimic with model in line 518.

Comment 10: Line 439. Reference 88 (Mariani, 2012) appears outdated for this context. Please include a more recent review on cortical development or insert Reference 89 in this location.

Response 10: Thank you for pointing this out. We have, accordingly, inserted reference 89 (now 126) in this location in line 525.

Comment 11: Lines 446–447. The claim that animal models fall short in capturing the complexity of human brain development should be rephrased for precision. Indeed they fall short in replicating cell diversity and human genetics.

Response 11: Thank you for pointing this out. We have, accordingly, rephrased the sentence and added references to emphasize this point between lines 554 and 558:

“This feature is particularly important for neurodevelopmental disorders, where animal models are limited in revealing some of the most fundamental aspects of development, genetics, pathology, and disease mechanisms that are unique to humans such as neocortical development, including cortical expansion, protracted time of development, and genetics (129,130)”.

This information is highlighted in yellow.

Comment 12: Line 459. A reference is missing. Please ensure this section is appropriately cited.

Response 12: Thank you for pointing this out. We have revised the text and added reference 124 in line.

By addressing these issues, the manuscript will present a more rigorous and coherent analysis of the literature.

Reviewer 3 Report

Comments and Suggestions for Authors

 Summary of the article

The article explores the impact of cannabinoids on human neurodevelopment using innovative bidimensional and tridimensional cultures derived from human pluripotent and induced pluripotent stem cells. It reviews the clinical evidence of the effects of cannabinoids during pregnancy and their role in altering neuronal differentiation and function. The study also highlights the importance of these models in understanding cannabinoid-related neurodevelopmental changes while emphasizing their potential for future research.

Comments:

1. All figures require detailed legends to explain and describe the content clearly.

2. Line 263: The subheading "2.2 Axon elongation and synaptogenesis" appears out of place as there are no other subheadings in this section.

3. It is recommended to include subheadings for each section to improve the article's structure and make it more organized and reader-friendly.

4. Before introducing human bidimensional and tridimensional cultures, it would be beneficial to discuss the limitations and disadvantages of current animal models for studying the effects of cannabinoids on neural development. This context would highlight the need for human-based models and strengthen the rationale for their use.

Author Response

REVIEWER 3

Comments: 

Comment 1: All figures require detailed legends to explain and describe the content clearly.

Response 1: We thank the reviewer for the important comments on this document. In attention to this point, in the corrected version of the draft we have included detailed legends to explain and describe the content in a clearer way.

Comment 2: Line 263: The subheading "2.2 Axon elongation and synaptogenesis" appears out of place as there are no other subheadings in this section.

Response 2: We thank the reviewer’s observation. We corrected this subheading and added further subheadings to provide a congruent and clearer organization of the manuscript.

Comment 3: It is recommended to include subheadings for each section to improve the article's structure and make it more organized and reader-friendly.

Response 3: We believe this is a valuable advice. In response to the reviewer’s request, we added subheadings to every section of the manuscript to provide a clearer organization of it. Subheadings are highlighted in yellow.

Comment 4: Before introducing human bidimensional and tridimensional cultures, it would be beneficial to discuss the limitations and disadvantages of current animal models for studying the effects of cannabinoids on neural development. This context would highlight the need for human-based models and strengthen the rationale for their use.

Response 4: We thank the reviewer for this valuable advice. Between lines 428 and 437 highlighted in yellow we have added the following information regarding the disadvantages of the current animal models for studying the effects of cannabinoids on neural development:

“Animal models have played a crucial role in expanding our understanding of how cannabinoids influence neural development. However, they are accompanied by several significant limitations. Key differences in the timing of brain development, variations in the components of the endocannabinoid system, and discrepancies in receptor distribution between humans and animals often constrain the relevance of these findings to human biology [42]. For instance, rodent models are unable to fully replicate the complexity of human neuro-development or the unique features of the human endocannabinoid system, making extrapolation of results to humans challenging [98,99]. Additionally, ethical considerations and the absence of a human-specific biological context further limit their applicability for studying prenatal cannabinoid exposure [100]”.

We further added the following paragraph to highlight the importance of using PSCs as a study  model to study the effect of cannabinoids during development (lines 448-454).

“To overcome these challenges, human-based models, including bidimensional and tridimensional neuronal cultures derived from PSCs, offer a more accurate alternative. These models enable researchers to study human neurodevelopment and the effects of cannabinoids in a controlled environment that closely mimics human biology [102]. By providing a framework that accounts for the intricacies of human neural development, these models offer a powerful tool to address the limitations inherent in animal and clinical studies”.

Round 2

Reviewer 2 Report

Comments and Suggestions for Authors

Dear Authors,

Thank you for your careful revisions of the manuscript. Your review effectively synthesizes both clinical evidence regarding cannabinoid consumption during pregnancy and emerging findings from human pluripotent stem cell models, providing a comprehensive overview of this important field.

I believe the manuscript is now suitable for publication. Thank you for your diligent work on this contribution.